# Evaluating an Adapted Physical Activity Program for University Students and Staff Living with a Physical Disability and/or Chronic Condition through a Self-Determination Theory Lens

Tayah M. Liska [1,*], Olivia L. Pastore [1], Gabrielle D. Bedard [1], Crystal Ceh [2], Leah Freilich [3], Rachel Desjourdy [3] and Shane N. Sweet [1,4]

1 Department of Kinesiology and Physical Education, McGill University, Montreal, QC H2W 1S4, Canada
2 McGill Athletics and Recreation, McGill University, Montreal, QC H2W 1S4, Canada
3 Student Accessibility and Achievement Office, McGill University, Montreal, QC H3A 2R7, Canada
4 Center for Interdisciplinary Research in Rehabilitation of Greater Montreal, Montreal, QC H3S 1M9, Canada
* Correspondence: tayah.liska@mail.mcgill.ca; Tel.: +1-613-983-3850

**Abstract:** The purpose of this mixed-method study was to (1) examine the effect of an adapted physical activity program, Fitness Access McGill (FAM), on leisure-time physical activity (LTPA), autonomous and controlled motivation, and the basic psychological needs of self-determination theory among university students/staff with a physical disability and/or chronic conditions, and (2) explore participants' experiences after completing FAM. Nineteen participants completed validated questionnaires for all study outcomes pre- and post-FAM. Nine participants partook in a 30–60 min semi-structured interview conducted within three months of completing FAM. Quantitative data were analyzed using repeated measures effect size calculations. Qualitative data were analyzed using directed content analysis. Participants reported an increase in total LTPA (dRMpooled = 0.58), with the greatest positive change on strenuous intensity (dRMpooled = 0.81). Large effects were found for changes in autonomous motivation (dRMpooled = 0.52), autonomy (dRMpooled = 0.79), competence (dRMpooled = 0.79), and relatedness (dRMpooled = 0.89). Participants reported FAM being supportive towards their psychological needs, the development of a LTPA routine, and enhanced overall well-being. Future research can be built upon this study to develop a robust understanding as to how need-supportive, adapted LTPA programs could be implemented within community settings or out-patient rehabilitation to support exercise engagement, physical health and overall well-being among adults with disabilities.

**Keywords:** physical exercise; fitness; social participation; quality of life

## 1. Introduction

Approximately 24% of first year university students and 30% of middle-year university students self-reported having disabilities in Canadian Universities [1,2]. In midwestern university settings, 15% (*n* = 138) of staff reported having disabilities, with mobility issues (40%) being the disability most frequently reported among staff [3]. The World Health Organization (WHO) defines *disability* as an umbrella term for impairments, activity limitations, and participation restrictions [4]. Among Canadians, the two most prevalent types of physical disabilities are mobility and flexibility restrictions [5]. Disabilities can also be associated with chronic conditions, including heart disease, respiratory disorders, and psychological conditions [6]. Individuals with physical disabilities and/or chronic conditions often face numerous barriers such as inaccessible faculties and infrastructure, limited accessible transportation and intrapersonal factors, hindering their ability to participate in society [7]. Despite efforts in creating an inclusive environment for individuals with disabilities, research and initiatives specifically for university students and staff with

disabilities in Canada remain limited. Thus, it is imperative to focus on this segment of the Canadian population to promote active participation in society.

One area that requires attention is the inadequate participation in leisure time physical activity (LTPA) among university students and staff with disabilities. LTPA is a type of physical activity individuals engage in during their free time (e.g., sports, exercising, walking/wheeling) [8]. Engagement in LTPA provides numerous benefits, including psychological and physical well-being [9–11]. Physical activity levels has also been associated with life satisfaction among individuals with disabilities [12]. However, researchers advocate that general physical activity guidelines, such as the WHO physical activity guidelines [4], do not include measures to assess LTPA in disability populations, preventing suitable recommendations of LTPA participation for adults with disabilities [13]. In the university setting, university students with physical disabilities use recreational facilities less than five times per semester [14]. Implementing LTPA initiatives grounded in behaviour change theories, however, has been proven to be conducive towards increasing LTPA participation among individuals with disabilities [7,15,16]. The inclusion of behaviour change theories can be highly valuable in promoting and encouraging LTPA participation among university students and staff with disabilities.

One meta-theory that has been found to be effective in the context of LTPA and disability is the self-determination theory (SDT) [15,16]. SDT focuses on the role of motivation and psychological needs in driving human behaviour [17]. A key tenet of SDT is that individuals have three basic psychological needs: *autonomy* (i.e., a sense of volition/freedom of choice), *competence* (i.e., a sense of belief in one's abilities), and *relatedness* (i.e., a sense of connection to others). When the three basic psychological needs are satisfied, individuals have increased *autonomous motivation*, which is driven by internal factors (e.g., interest, values, and satisfaction) [18]. Previous studies have shown that physical activity programs based on SDT have been effective for increasing SDT variables and physical activity behaviour for individuals living with a disability [16–19]. To date, however, there remains limited understanding as to whether campus-based adapted physical activity programs for university students and staff with physical disability and/or chronic conditions are effective for increasing individuals' LTPA and SDT variables.

To support LTPA participation for university students and staff living with a physical disability and/or chronic condition, a campus-based adapted physical activity program called Fitness Access McGill (FAM) was co-developed. FAM is grounded in SDT and offers personalized adapted physical activities to help individuals with a physical disability and/or chronic conditions on campus engage in LTPA over a 10-week period. Understanding the effect of FAM on participants is important as it can provide insight into how FAM can fulfill the basic psychological needs of SDT and foster LTPA engagement among students and staff with disabilities or chronic conditions.

The purpose of this mixed-method study was twofold: (1) to examine the effect of an adapted physical activity program, FAM, on LTPA, autonomous and controlled motivation, and the basic psychological needs of SDT (i.e., autonomy, competence, relatedness) among students and staff with a physical disability and/or chronic condition, and (2) to explore the experiences of FAM participants after completing the program. For the quantitative portion of the study, we hypothesized that FAM participants would report an increase in physical activity participation, autonomous motivation, and satisfaction of the psychological needs after the program compared with pre-program [16,19,20]. There was no hypothesis for research question 2 as it focused on describing experiences in FAM.

## 2. Materials and Methods

### 2.1. Research Design and Paradigm

The research team followed a pragmatic paradigm which posits that researchers should use the philosophical and methodological approaches that are most appropriate to answer the research questions [21]. Pragmatism acknowledges that meaning is socially constructed with an individual's past and current experiences and that there is no single

truth to uncover [21]. Pragmatism is the appropriate epistemology for this study as it allows the researchers to explore a phenomenon using different methodological approaches. As such, we used a mixed-method convergent parallel design to answer our first research question [22]. Using both quantitative and qualitative data (i.e., triangulation) can strengthen our conclusions regarding the FAM program [23]. For the quantitative data, we used a pre-post design whereby data were collected before and immediately after the completion of the FAM program. To obtain the qualitative data, we used a generic qualitative design and used semi-structured interviews up to three months after the completion of the FAM program. To answer research question 2, we used the qualitative data to explore the participants' experiences of FAM and to understand how FAM may have provided adequate support towards the participants' LTPA engagement, motivation, and psychological needs.

### 2.2. Fitness Access McGill (FAM)

FAM is a 10-week adapted physical activity program co-developed by McGill's Student Achievement and Accessibility Office, Athletics and Recreation, and the Department of Kinesiology and Physical Education. FAM received research ethics approval from McGill University's research ethics board (approval #111-0818). Upon providing informed consent for enrollment, participants undertook a fitness evaluation (in week 1) and were provided a personalized exercise prescription (in week 2) by a certified kinesiologist. See Supplemental Materials for the fitness evaluation procedure and an example of an exercise prescription, respectively. For the rest of FAM, the participants were placed in groups of 2 or 3 and engaged in drop-in exercise sessions twice a week for 60 min, which were supervised by upper year undergraduate kinesiology internship students. During the drop-in sessions, each participant engaged in their prescribed exercises using their body weight or the adapted equipment in the FAM gym on campus. Though participants engaged in their individual exercises and received personalized feedback from the interns, the interns also created a social space for participants to exercise and converse with each other.

The kinesiology interns took undergraduate courses focused on kinesiology topics (e.g., exercise and health psychology, adapted physical activity) and received training in SDT prior to starting the program. Additionally, the interns gained experiential practice by shadowing a certified kinesiologist during the initial assessments and exercise prescription. The interns also received a resource booklet prior to starting FAM, which outlined the various physical disabilities and/or chronic conditions (e.g., what they are and recommendations for exercises related to the disability and/or chronic condition), person-centered approaches to foster autonomous motivation, and behaviour change techniques.

Due to the COVID-19 pandemic, this study reports data from two iterations of FAM. Before the COVID-19 pandemic (September 2019–March 2020), the assessment and group sessions occurred in-person on campus. For the first two weeks of group exercise sessions, the participants came in-person to a private adapted laboratory. For the remainder of the sessions, participants were transitioned from the adapted laboratory to the McGill Fitness Centre as they continued to be supervised by the kinesiology internship students. During the COVID-19 pandemic (September 2020–March 2021) FAM transitioned to an online format. Due to public health restrictions, the assessments and group exercise sessions occurred online through Microsoft Teams.

### 2.3. Participants and Procedures

Students enrolled with the Student Achievement and Accessibility Office were invited to participate in FAM through an email invitation. Additionally, FAM was promoted through university websites and newsletters to recruit staff members living with a physical disability and/or chronic condition. To be included in this study, participants must have been a student or staff member with the university, be living with a physical disability and/or chronic condition, be English or French speaking, have no cognitive/memory impairments, and must have completed the full FAM program. Once participants provided informed consent, they filled out a demographic questionnaire and were sent a link for an online survey to assess

baseline levels of LTPA, autonomous and controlled motivation, and the psychological needs (i.e., autonomy, competence, and relatedness). Next, the participants underwent the 10-week FAM program, during which no data collection took place. After completion of FAM, the participants filled out a post-program survey to assess LTPA, autonomous and controlled motivation, and the psychological needs. Lastly, the participants engaged in a semi-structured interview post-program (M = 28:45 min SD= 5:26 min).

### 2.3.1. Quantitative Measures

The participants completed validated and reliable questionnaires at baseline and post-program assessing LTPA, autonomous and controlled motivation, and the psychological needs of autonomy, competence, and relatedness. To examine LTPA, we used a modified version of the Godin Leisure Time Exercise Questionnaire, which assesses total LTPA as well as strenuous, moderate, and mild intensity levels [24]. For each intensity of exercise, the questionnaire asks participants "*Considering a typical week in the past 2 months, how many times on average have you done (intensity) exercise for more than 20 min during your free time (write the appropriate number of times per week)*" followed by "*For approximately how many minutes do you participate in each (intensity) exercise session?*" To obtain a separate weighted summed score for each intensity (also known as the Leisure Score Index or LSI), the frequency was multiplied by its respective metabolic equivalent task value [i.e., ($5 \times$ moderate LTPA): ($9 \times$ strenuous LTPA): ($3 \times$ mild LTPA)]. A total physical activity score (i.e., total LTPA) was calculated at each timepoint by totaling the intensity LSI values, as recommended by Godin (2011). The average and standard deviation of these values is reported for all participants at each timepoint (Table 1). As there are no guidelines and recommendations that are specific to individuals with a range of physical disabilities and/or chronic conditions, we did not use any cutoff number to indicate participants as insufficiently active or not.

**Table 1.** Means (Standard Deviations) for study outcomes.

| Variable | $M_{Baseline}$ (SD) | $M_{Endpoint}$ (SD) | *t*-Value | *p*-Value |
|---|---|---|---|---|
| Total LTPA [a] | 184.00 (109.48) | 248.45 (122.15) | 2.51 | 0.02 |
| Strenuous LTPA [a] | 27.37 (38.31) | 63.42 (56.23) | 3.30 | 0.003 |
| Moderate LTPA [a] | 62.42 (72.44) | 85.55 (67.07) | 1.16 | 0.26 |
| Mild LTPA [a] | 94.21 (82.95) | 99.47 (98.08) | 0.26 | 0.79 |
| Autonomous Motivation [b] | 37.84 (3.93) | 39.40 (2.87) | 2.00 | 0.06 |
| Controlled Motivation [b] | 39.32 (15.87) | 40.74 (10.20) | 0.35 | 0.73 |
| Autonomy [c] | 27.47 (5.64) | 31.33 (4.56) | 3.42 | 0.003 |
| Competence [c] | 21.93 (8.39) | 28.00 (7.32) | 3.40 | 0.003 |
| Relatedness [c] | 36.73 (11.29) | 45.33 (10.79) | 3.86 | 0.001 |

Note: LTPA, Leisure Time Physical Activity; SD, Standard Deviation. [a] Calculated from the Godin Leisure Time Exercise Questionnaire [24]. [b] Calculated from the Self-Regulation Questionnaire for Exercise [25]. [c] Calculated from the Psychological Need Satisfaction in Exercise Scale [26].

For motivation, we used the Treatment Self-Regulation Questionnaire for Exercise [25]. This questionnaire (based on self-determination theory) uses a 7-point Likert scale from 1 (Not true at all) to 7 (Very true) and prompts participants to respond to the question "*Why do you exercise?*", with 15 corresponding items related to the participants' reasons for engaging in or changing a health behaviour. Autonomous motivation, which makes up six of the fifteen questions from the questionnaire, indicates that an individual can have three types of motivation for exercise (two being extrinsic motivation and one being intrinsic). Under extrinsic motivation, individuals can have identified regulation (i.e., when exercise is positively endorsed and valued by the individual) or integrated regulation (i.e., when exercise is perceived as being part of the larger self). Individuals who have intrinsic motivation indicate that exercise is engaged in for its own sake, simply for the

pleasure, interest and satisfaction derived from performing it. Controlled motivation, which makes up six of the fifteen questions from the questionnaire, comprises two types of extrinsic motivation, including external regulation (i.e., exercise that is performed in order to obtain a reward or to avoid negative consequences) and introjected regulation (i.e., exercise that has been partially taken in by the person and is performed to avoid feeling guilty or ego-involved). Additionally, three of fifteen questions correspond to amotivation, which describes when an individual has an absence of motivation to engage in exercise. To calculate total values for each type of motivation (at separate timepoints), the items corresponding to autonomous and controlled motivation were individually summed [25]. The average and standard deviation of these values is reported for all participants at each timepoint (Table 1). We decided not to include amotivation in our overall analyses as all participants reported some type of motivation for exercise (i.e., autonomous or controlled). Overall, autonomous motivation is associated with positive health, behavioural, and psychological outcomes. In contrast, controlled motivation is linked to poorer health and well-being.

Lastly, we used the Psychological Need Satisfaction in Exercise Scale [26], which is an 18-item questionnaire used to assess changes in the psychological needs of self-determination theory (i.e., autonomy, competence, and relatedness). The participants were asked to rate each item on a scale of 1 (False) to 6 (True). Each question relates to how the participant feels during exercise regarding autonomy (e.g., *I feel free to exercise my own way*), competence (e.g., *I feel that I am able to complete exercises that are personally challenging*), and relatedness (e.g., *I feel like I share a common bond with people who are important to me when we exercise together*), with each need corresponding to six questions of the questionnaire. To calculate total values for each psychological need (at separate timepoints), the items corresponding to autonomy, competence, and relatedness were individually summed. The average and standard deviation of these values are reported for all participants at each timepoint (Table 1).

2.3.2. Qualitative Interviews

Each member enrolled in FAM was invited to participate in a one-on-one semi-structured interview up to three months after completion of the FAM program. While some participants participated in the interview immediately after completion, the overall interview timeline varied due to the scheduling constraints of participants. The post-program interviews were used to assess research questions 1 and 2, to examine the influence of FAM on LTPA, autonomous and controlled motivation, and the basic psychological needs of SDT among students and staff with a physical disability and/or chronic condition and to explore the experiences of FAM participants after completing the program. An interview guide was created for the interview, asking participants about their LTPA, autonomous and controlled motivation, and the basic psychological needs of SDT in the context of FAM, as well as their experiences engaging in the program. Examples of interview questions asked include, *"how have you felt since the end of the FAM program"* and *"did the exercise programs and modifications provided by the kinesiologist support your physical activity"*. In addition to asking the participants about their LTPA participation within FAM, the interviews also allowed us to explore how the program potentially influenced their LTPA participation outside of FAM. All interviews were conducted by video-call via Microsoft Teams and transcribed verbatim for analysis.

*2.4. Data Analysis*

2.4.1. Quantitative Analysis

Preliminary visual analyses to ensure normality were conducted using SPSS Statistical Software [27]. Given the smaller sample size, the effect size (rather than significance) was focused on to determine if individuals had changes in the outcome variables. Effect sizes using the Morris and DeShon [28] procedure for single-group pre-post designs were calculated for each of the outcome variables. Rather than using the correlation between pre-

and post-values, this procedure uses the standard deviation of the pre-test as this value is not influenced by the intervention. The effect size we chose to report ($d_{RM,pool}$) uses the pooled standard deviation, controlling for the intercorrelation of both groups. According to Cohen [29], the following guidelines were used to assess effect sizes and strength of the relationships: $d_{rm}$ = 0.10 to 0.29 (small); $d_{rm}$ = 0.30 to 0.49 (medium); and $d_{rm}$ = 0.50 to 1.0 (large). Paired-sample *t*-tests were still conducted to provide statistical trends and compliment effect size interpretations.

2.4.2. Qualitative Analysis

We employed a directed content analysis, using a deductive approach to organize and analyze the qualitative data. The method used prior theory to extend knowledge about a phenomenon [30]. Co-authors Tayah M. Liska (TML) and Olivia L. Pastore (OLP) deductively coded the data based on the psychological needs of SDT (i.e., autonomy, competence, and relatedness) and the other study variables (i.e., LTPA, autonomous and controlled motivation). After familiarizing themselves with the data, TML and OLP separately highlighted and coded the data using the SDT psychological needs and other study variables as predetermined themes, where possible [30]. Any text that did not fit within a predetermined theme was given another inductively coded label [30]. Next, TML and OLP examined the pre-determined themes to determine whether sub-themes were needed to provide further grouping of the coded data. After, the coded data were organized into the predetermined themes, and the coded text items within established sub-themes were organized into the overarching psychological need or other study variable themes. Items of text given another label were inductively coded and organized into separate themes.

Rigor and trustworthiness. Data-source triangulation and multiple-analyst triangulation of the data occurred to ensure validity of the data [31]. To ensure the rigor of the coded data, as well as consistency in the selection and organization of codes across the researchers, TML and OLP met six times throughout the coding process. After coding an initial number of transcripts independently, TML and OLP met to compare codes. When differences in the coded data appeared in the cross-coding comparison, both researchers discussed the difference and rectified the discrepancy. Independent and cross-coding comparison occurred for the remaining number of transcripts. After all the transcripts were coded, the researchers met again to compare their codes. Additionally, once the final codes were created, the researchers presented them to a 'critical friend', Shane N. Sweet (SNS) who has extensive knowledge of the FAM program and SDT variables. This individual was able to critique the relevance of the coded data to the predetermined SDT categories and additional SDT variables.

## 3. Results

### 3.1. Participants

Nineteen students and staff who were living with a physical disability and/or chronic condition, were English or French speaking and had no cognitive/memory impairments completed the FAM program (see Table 2). The physical disabilities and/or chronic conditions (some participants (21%) had more than one physical disability or chronic condition) of the participants included ACL injury (*n* = 1), amputation (*n* = 1), bulging disks (*n* = 1), chronic pain (*n* = 2), fibromyalgia (*n* = 4), flat feet (*n* = 1), multiple sclerosis (*n* = 1), osteoarthritis (*n* = 1), polyarthritis (*n* = 1), scoliosis (*n* = 1), severe asthma (*n* = 1), spinal stenosis (*n* = 1), spondylolisthesis (*n* = 1), stroke (*n* = 1), tendonitis (*n* = 4), and traumatic brain injury (*n* = 1). No participants required physical assistance to move around the community.

**Table 2.** Demographic variables for Iterations 1 and 2 of FAM (*N* = 19).

| Variable | Iteration 1: In-Person (*n* = 9) | | Iteration 2: Online (*n* = 10) | |
|---|---|---|---|---|
| | Quantitative (*n* = 9) | Qualitative (*n* = 4) | Quantitative (*n* = 10) | Qualitative (*n* = 5) |
| **Age** Mean (standard deviation) | 26.89 (5.88) | 26.29 (5.28) | 36.20 (15.41) | 36.20 (15.41) |
| **Gender** Female Male | 9 (100%) 0 (0%) | 4 (100%) 0 (0%) | 7 (70%) 3 (30%) | 4 (80%) 1 (10%) |
| **Status (*n*, %)** Staff Student | 0 (0%) 0 (0%) 9 (100%) | 0 (0%) 4 (100%) | 6 (60%) 4 (40%) | 3 (60%) 2 (40%) |

*3.2. Research Question 1: The Effect of FAM on LTPA, Motivation, and Psychological Needs*

Based on SDT, we provide data for each study variable. All names mentioned in the qualitative results are pseudonyms. All the quantitative results are reported in Table 1. Given the mixed-methods approach to this study, quantitative and qualitative results were integrated to demonstrate a measured effect and the participants' experiences associated with the predetermined and additional identified themes.

3.2.1. LTPA

Repeated measures effect size calculations revealed that the participants reported an increase in total LTPA from baseline to endpoint ($d_{RM,pool}$ = 0.58), with the greatest positive change in strenuous intensity ($d_{RM,pool}$ = 0.81), followed by mild and moderate ($d_{RM,pool}$ = 0.27, $d_{RM,pool}$ = 0.06, respectively). The qualitative analysis supported these quantitative findings. For example, Alex, a staff member, spoke about how their LTPA "changed quite a bit" from FAM and that 90% of the changes, if not all, were due to the program. The qualitative component also allowed us to understand that the routine developed from participating in weekly FAM sessions may explain this increase in LTPA. The participants reported that due to attending FAM, they were more aware of regularly scheduling LTPA into their daily life, ensuring consistency in their LTPA participation. Furthermore, the participants emphasized that they developed an awareness that they could engage in LTPA while living with their disability, as well as maintain the physical and psychological benefits gained from establishing an LTPA routine. As one participant stated: *"[FAM] definitely helped to develop a different perspective of physical activity, maintaining a workout routine of some sort with my illness"* (Alex, staff member).

Participants also described their continued engagement in LTPA within three months after the completion of FAM. For instance, they continued to engage in various forms and intensities of LTPA after FAM including running, walking, yoga, or continuing their exercise prescription. For example, Carey (student) reported that they were *"continuing to do their [FAM] exercises"*, whereas Lenny (student) explained *"I'm running like once every two days so about like three or four days a week. I've been doing something like 3–4 k every two days"*, and Robbie (staff member) opted for walking: *"I'm not doing a lot of intensity but I'm doing more like moderate exercise like walking"*.

3.2.2. Autonomous and Controlled Motivation

There was a large effect for improvements in autonomous motivation in the quantitative results ($d_{RM,pool}$ = 0.52). The participants also expressed in the interviews that FAM supported their autonomous motivation for LTPA as the program reaffirmed familiar feelings and knowledge about previous engagement in LTPA or sports. For example, Jessie, a student in FAM, explained that:

*"[FAM] kind of brought back the reason I like doing sports and working out in the first place, because it made me feel good not just to get fit. Getting fit is obviously part of it but I just like having that energy and I like that I wanted to go back. I feel that's what surprised me a lot."*

Participants spoke of how FAM reaffirmed familiar, or once present, positive feelings and actions towards engagement in LTPA. The feelings of increased energy and fitness were a positive, motivational factor in why they chose to go back to participate in physical activity.

The participants also expressed an increase in controlled motivation and felt a sense of accountability to attend FAM sessions and complete their prescribed exercises in their own time. Consistent with these findings—however inconsistent with our hypothesis—there was a small effect for increases in controlled motivation ($d_{RM,pool}$ = 0.09) from baseline to post-program. This could be because once participants were in the FAM sessions, they also described an inclination to "go further" and "do more" with their exercises. This sense of accountability was reported to be from two sources within FAM: (1) having FAM scheduled in their calendar, and (2) not wanting to disappoint the FAM interns by not showing up or giving effort. The overall increase in autonomous and controlled motivation could be due to a perceived increased in the three basic psychological needs of autonomy, competence, and relatedness.

### 3.2.3. Psychological Need Support and Satisfaction

Autonomy. There were increases in the participants' autonomy through FAM, as seen in both the qualitative and quantitative findings. Specifically, there was a large effect size for increases in autonomy from baseline to post-program ($d_{RM,pool}$ = 0.79). The increased sense of autonomy experienced by the FAM participants was due to the kinesiologist and kinesiology interns listening to their needs and adapting the program appropriately. For example, a participant stated that "There were some points where I decided that a certain exercise wasn't really working for me, my body and my needs", which demonstrates a sense of autonomy in being able to change their program. This participant continued by explaining "they [kinesiology interns] were super quick to offer alternatives, adjustments, or modifications that we could make so they were very supportive and flexible, and just really a huge help in that regard" (Alex, staff member). These findings lead to the interpretation that the kinesiologist and kinesiologist interns used autonomy support strategies (e.g., offering alternatives) during program sessions, which allowed for fostering of autonomy satisfaction within the participants.

Competence. There was also a large effect for increased competence from baseline to post-program ($d_{RM,pool}$ = 0.79). The responses provided in the interviews supported the interpretation that participants had increased competence and comfort to engage in LTPA individually and outside of the FAM sessions. Overall, this increase in competence likely contributed to continued LTPA for some participants who were interviewed at three-months follow-up: "Now that I have a little bit more confidence in what I'm doing and also now that I do think I'm definitely more comfortable than I was exercising on my own." (Jessie, student). The reported increase in participant competence could be due to the high competence support provided by the kinesiology interns. This is supported by the participants' reports that the kinesiology interns provided attention, individual- and exercise-specific feedback, and explanations of how to perform exercises suitable for their individual needs. The in-person demonstrations and corrections of exercises appeared to be helpful to remove any sense of uncertainty: "I think what [the interns] were most helpful with was correcting some of the ways I was doing things" (Avery, staff member). Even though the participants felt more confident performing LTPA on their own, a large component of FAM described in the interviews was a sense of relatedness to the kinesiology interns and other FAM participants.

Relatedness. With the strongest effect size ($d_{RM,pool}$ = 0.89), the participants reported an increase in their sense of relatedness from FAM due to feelings of inclusivity and understanding while attending FAM sessions:

*"Having other people who have similar issues was a really positive aspect for me because I felt like they really understood and instead of feeling embarrassed that I couldn't do things I felt proud that I was able to do it with other people and proud to see that they were doing it too."* (Jessie, student).

Participants elaborated that FAM created a sense of relatedness and a supportive environment whereby they could engage in LTPA with other adults living with physical disabilities. In addition to feeling connected with peers, participants highlighted that the kinesiology interns promoted a sense of relatedness by creating a friendly, encouraging and welcoming environment through on-going verbal encouragement and being warm and caring during each session. The welcoming and inclusive environment helped participant feel they belonged while creating a sense of community: *"I feel like I learned that exercise and just being with people it's more than just working out, it's a community and it gives you something to look forward to even on a day when you're not feeling great"* (Jamie, student).

### 3.3. Research Question 2: Exploring the Experiences of FAM Participants

Beyond the psychological needs, motivation, and physical activity, participants emphasized that attending FAM improved their psychological well-being, participation in daily and social activities, and physical well-being. They also provided feedback on the program and program delivery options. Given the mixed-methods approach to this study, quantitative and qualitative results were integrated to demonstrate a measured effect and participants' experiences associated with the predetermined and additional identified themes.

#### 3.3.1. Psychological Well-Being

Despite academic, professional, and other responsibilities (e.g., extracurricular commitments, family responsibilities) that contributed to the participants' busy schedules, there was consensus that engagement in the FAM program and subsequent LTPA outside of the FAM sessions enhanced their psychological well-being by allotting *time for oneself* and *increasing happiness*.

Time for Oneself. Participants reported that among their multiple responsibilities and busy personal schedules, having a designated time to engage in LTPA reinforced the importance of designating time to themselves within their day. Interestingly, participants had an increased awareness that engaging in LTPA served as a beneficial means of respite within their daily schedules. This important awareness described by participants supported the interpretation of the data and the overall importance of providing time for oneself.

*"Overall, I think [FAM] helped me to make that time for myself and my body, my health, my well-being, [to] make that time more of a priority and taking it more seriously. [It] hugely impacted my mental and emotional health as well as my work-life ratio, that's been really big"* (Alex, staff member).

Increased Happiness. There was a collective sense that participation in the FAM program provided participants a boost in overall mental health and mood, leading to increased happiness: *"I think [FAM] might have made me a little bit happier because I really like exercising"* (Carey, student). The sense of pleasure, fulfillment, and accomplishment described by participants from sustained engagement in LTPA initiated by the FAM program stimulated awareness of the interpersonal benefits of LTPA participation and an increased intrinsic motivation for participants to continue to be active:

*"...now it's like "No I have to [do LTPA], and I will feel better, and I will be more productive and happier" and the more I do [LTPA] the more I feel inspired to do it and the more I want to do it so therefore the better it is for me all around"* (Alex, staff member).

#### 3.3.2. Participation in Daily/Social Activities

The participants also spoke about an increased sense of social well-being from FAM, specifically through having increased ability, ease, and comfort in participating in daily and

social activities. For example, routine daily activities (e.g., cooking, cleaning) that were once considered too strenuous by participants became much more manageable: *"[FAM] helped me practically. I told one of the interns that for a few months I never cook with this certain pot because it's just so heavy"* (Rene, student). Other participants discussed developing communication skills in the smaller group setting of FAM which led to being more comfortable talking louder in larger social settings. Furthermore, participants discussed about feeling more comfortable participating in the university's fitness center. As Jessie, a student, explained:

> *"...a big one is just feeling like I don't belong there [at the fitness centre]...I was very nervous about it like it's definitely super intimidating...like feeling like other people are really fit and they know what they're doing, and I feel like I'm clueless. Now that I have a little bit more confidence in what I'm doing and also now that I understand that that's not true and basically most people who are there are just worried about what they're doing themselves and not actually judging what I'm doing."*

### 3.3.3. Physical Well-Being

The participants reported an increase in physical function and capability as a result of engaging in the FAM program. They noticed an *increased physical capability* through improved energy, strength, and flexibility when engaging in LTPA, exercise, and completing daily tasks. Also, participants reported *improved pain management* because of on-going LTPA participation through FAM, thus supporting their overall physical well-being.

Increased Energy. Participants described the encouragement they felt when recognizing that their routine engagement in LTPA supported improvements in energy and functionality that, prior to FAM, remained weak or depleted:

> *"When I first started FAM I had spent almost two years in bed because of my illness and I was so weak, my arm was weak from brushing my teeth and I had no energy whatsoever...and in doing FAM for a few semesters I actually started doing stuff like running"* (Rene, student).

The participants spoke of how their disability or illness represented limitations towards engaging in LTPA, such as low energy levels or reduced physical strength. However, after receiving tailored support within the accessible environment of FAM, participants noticed gains in their strength, flexibility, and energy to engage in various LTPA or exercises: *"I have more energy, I sleep better and well I'm thinking immediately after finishing the program"* (Rene, student), and *"I saw even after the first couple weeks I was able to lift way more weights. In a good way, it made me feel good and I just felt much stronger"* (Jamie, student).

Improved Pain Management. A disability-specific outcome reported by some participants was the reduction in pain from engaging in the FAM program, and participants increasing their adoption of LTPA as a means of pain management. Participants spoke of how they began to recognize how LTPA and exercise can indeed be supportive of pain management and be an additional means of mitigating short-term and long-term occurrences of pain: *"[FAM] improved my outlook on my pain because it made me realize that if I'm in pain I can just workout and after a couple of weeks or three weeks it goes away"* (Charlie, student). In addition to the positive psychological awareness of the occurrences of pain, participants spoke of how they became increasingly aware of their bodies and changes in their pain as a result of participating, or not participating, in LTPA: *"I definitely learned about my body. That there were exercises that were helpful or managing my pain"* (Rene, student). Although LTPA is not a replacement for methods of pain management nor a method for providing pain management for all persons with a physical disability and/or chronic condition, the perceived benefit LTPA can provide for some individuals for pain management is encouraging.

### 3.3.4. Program Delivery

Within the interview responses, participants alluded to different advantages and disadvantages of the in-person versus online delivery. Participants who engaged in the online iteration of FAM highlighted that it provided additional accessibility and accommodation

for their disability. First, the online version was more accessible to some participants because the in-person fitness facility was in a building on top of a hill, which made it more challenging to travel to and attend in-person sessions. Also, having the online option made the FAM program psychologically accessible as participants reported they could engage in the exercises at a pace that was suitable for them without feeling pressure to keep up with a group:

> "*Normally with the group class [FAM] challenging for me because I can't always do what everyone else is doing but I'm finding being at home and there's no camera on me I can—I feel free to sit out for a lot of the... exercises that are too much*" (Max, student).

Even though the online version was accessible in many ways, participants discussed that having the in-person option allowed for critical social connection and tangible exercise-specific feedback from the kinesiology internship students. Having the direct feedback regarding their engagement in exercise was perceived as highly valuable to the participants in supporting their competence to effectively engage in the exercise, as previously discussed. Overall, as participants saw value in both the online and in-person iterations of FAM, they recommended having a hybrid program whereby they have the choice of attending sessions online or in-person.

## 4. Discussion

The results of this study provide insight into the benefits of participating in an adapted physical activity program for university students and staff living with a physical disability and chronic condition. To address a call to provide more local accessible physical activity programs for adults living with disabilities [32], this study aimed to capture insights about participants' experiences while enrolled in the FAM program to determine if a need-supportive environment provided adequate guidance and support towards participants' LTPA engagement, motivation, and psychological needs (i.e., autonomy, competence, relatedness). Both quantitative and qualitative results demonstrated support for FAM.

Our results indicated that FAM is conducive towards increasing LTPA participation, particularly LTPA after program delivery. This finding is consistent with previous evidence, which showed that engagement in LTPA in inclusive, community-based LTPA programs supported increased LTPA participation [12]. Even though our participants were engaging in more mild and moderate forms of LTPA after FAM, the largest increase was seen for strenuous intensities. This finding could be because participants mentioned continuing to do their prescribed FAM exercises, which were incremental intensities and more strenuous in nature (e.g., walking or jogging longer distances, sit to stand, bicep curl, countertop push-up). The participants' improvements in strenuous LTPA are promising, as evidence indicates that moderate to strenuous forms of LTPA are associated with optimal health and fitness benefits in young and middle-aged adults living with a physical disability [33].

Moreover, the need-supportive environment of the FAM program likely supported the participants' satisfaction of autonomy, relatedness and competence, increasing their motivation and LTPA, which is consistent with recent reviews of SDT-based interventions [20,34]. Also, LTPA programs grounded in SDT and tailored to a disability context have been shown to be conducive towards increasing autonomous motivation and LTPA participation [15,16]. Such increases in SDT variables and LTPA participation can be attributed to the kinesiologist and kinesiology interns in FAM who created a welcoming and inclusive environment and provided personalized feedback, hence providing relatedness support. These results are similar to an implementation evaluation by Rocchi et al. (2021) [35] reporting that participants with spinal cord injury who were more engaged during their LTPA sessions and had more discussion with their exercise counselor reported an increase in motivation and LTPA upon completion of their program. The continuous interaction between participants and their counsellor demonstrates the key role engagement has in the success of need-supportive LTPA interventions [35]. Also, peer support has been identified as being supportive and motivating towards the participation of adults with disabilities in LTPA programs [36]. In the structured context of peer mentorship, those who receive mentorship

report increased feelings of confidence and mutual understanding from peers [37]. The on-going support received from peer interactions in LTPA programs can allow for large increase in relatedness, supporting individuals' psychological needs, promotion of LTPA, and enhanced well-being.

The participants also reported higher levels of autonomy and competence after completing FAM due to the guidance, personalized programs and individualized feedback provided by this program. Previous research has indicated that personalized feedback or tailoring LTPA programs to the individual supports the individual's autonomy and competence towards LTPA participation [15]. The presence of interpersonal behaviours by LTPA counsellors or kinesiologists in LTPA programs has been suggested to provide the necessary support to foster long-term LTPA [15]. The increased levels of autonomy and competence are especially important for individuals with disabilities, as intrapersonal barriers (e.g., psychological factors, pain, fatigue) and interpersonal barriers (e.g., social support, societal attitudes) towards LTPA participation predominate among individuals with disabilities [38]. The inclusion of SDT within LTPA interventions has been identified as being an effective LTPA-enhancing strategy for adults with disabilities [38]. The psychological needs within SDT provide theoretical relevance at the interpersonal level, and thus, can serve as a means of interpersonal support and guidance in LTPA settings for adults with disabilities [38].

According to SDT, once the psychological needs are satisfied, there is an increase in autonomous motivation and a decrease in controlled motivation [17–34]. Our findings confirm the former such that participants reported an increase in autonomous motivation for LTPA after engaging in FAM. The results of this study echo previous findings exploring need-supportive LTPA programs for adults with physical disabilities [39]. Bremer et al. (2022) [15] reports LTPA counselling sessions for adults with disabilities increased participants' autonomy, competence, and relatedness towards LTPA, leading to increased levels of autonomous motivation and competence/self-efficacy towards engaging in LTPA. Taken together, when adapted LTPA programs for people with disabilities ensure the presence of a welcoming environment, are tailored to an individual's capability, and provide personalized feedback, there is greater autonomous motivation and sustainability towards LTPA participation [12,15,16].

Importantly, the findings of this study also demonstrate that after engaging in the program, the participants reflected on their psychological and physical well-being gains resulting from FAM. These results are rather intriguing, given that the benefits from engaging in a LTPA program extend beyond psychological needs satisfaction and motivation. Although the outcome of well-being was coded separately from SDT variables, psychological well-being is promoted when an individual's psychological needs are satisfied [18]. Therefore, our results further align with SDT in that the satisfaction of the participants' psychological needs during the FAM program could have inherently increased feelings of well-being. Previous tele-health interventions grounded in SDT have demonstrated improvements in participants' overall life satisfaction and generated feelings of more meaningful life engagement [16]. Given the intrapersonal benefits experienced by participants that extended beyond their psychological needs during the FAM program, LTPA programs that are tailored to support psychological needs may also support a more robust sense of well-being. The mixed-methods approach to this study allowed us to consider the participants' entire experience of FAM, allowing us to extend our findings beyond the original focus on SDT and LTPA variables. This holistic approach can further support the effectiveness of behaviour change interventions [40].

Moreover, evidence indicated that community-based LTPA programs can support continued engagement of adults with disabilities in LTPA, as well as their engagement in social activities [7,41]. Community-based LTPA programs are conducive to supporting social inclusion and reducing feelings of social isolation, while improving LTPA and participation in daily and social activities among adults with disabilities [41,42]. When the psychological needs of competence and relatedness are enhanced through social interaction within community-based LTPA programs, there is an increase in quality of life and social

participation [43]. However, researchers identified reductions in participants' social interactions the longer their participation in the program had become due to increased feelings of comfort and competence in engaging in their exercise independently [12]. Therefore, increasing feelings of competence and relatedness can be act as mediators between LTPA and quality of life, participation in social activities, and confidence towards engagement in LTPA among adults with disabilities [12,43].

The evident improvements in physical health outcomes and social interactions from LTPA participation support the call to promote equitable LTPA participation and programs for adults with physical disabilities [44]. The participants had improved physical capability and energy levels while engaging in tasks of daily living, improvements in pain management and overall fitness, highlighting the health benefits retained by adults with physical disabilities from engaging in LTPA. Strategies for promoting LTPA, such as individually tailored exercise programs, counselling support, and regular follow-ups with individuals within the program, have been previously shown to support improved LTPA engagement and health outcomes among adults with physical disabilities [15–44].

Moreover, the integration of behaviour change techniques and a need-supportive environment into LTPA programs has been identified as conducive to increasing continued LTPA engagement by participants outside of an intervention setting [40]. Research has suggested that LTPA interventions that primarily utilize behaviour change techniques from behaviour change theory result in a high likelihood of participants maintaining their LTPA upon completion of the intervention [45,46]. However, limited research has been conducted exploring how improvements in LTPA are maintained following an intervention [46]. It is evident that adults with disabilities who continue to engage in LTPA experience positive health outcomes. However, there is limited understanding about what specifically supports LTPA maintenance for adults with physical disabilities. Therefore, further research in LTPA maintenance is warranted.

The online option provided increased flexibility and choice for participants. Having the choice to attend FAM sessions online was seen as advantages as it could accommodate physical and psychological accessibilities issues. For instance, online sessions reduced the need for difficult travel to in-person sessions while allowing participants to exercise at a personalized pace and level of intensity. The online option for attending sessions supports increasing calls within the literature to use barrier-free methods for program engagement or dissemination to allow for equitable participation for adults with disabilities [44,47,48]. Decreasing barriers by using accessible means of physical activity program delivery, such as online delivery, is supportive of promoting physical activity among disability populations [32,48]. The enhancement of LTPA participation among adults with disabilities when access to high quality, adapted physical activity programs is available speaks to the continued need to implement programs that are universal in design and tailored to populations' needs [44,49]. The results of this study demonstrate that online delivery of LTPA programs is useful for adults with disabilities and is an effective means of enhancing their LTPA participation.

Previous literature shows that tele-health or virtual delivery of SDT-based LTPA interventions can be feasible and support increased LTPA participation [16,35]. Therefore, future LTPA programs should consider virtual hybrid approaches, as in this study, to enable increased accessibility and feasibility to support participation and continued engagement in LTPA among adults with disabilities.

## 5. Limitations and Future Research

This study was a mixed-method, quasi-experimental design that consisted of a small sample size with no control group for comparison. This design poses limitations for our findings as we cannot say for certain whether the results were due to FAM or if additional variables contributed to the study outcomes. Future studies should consider using a randomized control design with a larger sample size to determine cause and effect. Despite the smaller sample, our qualitative findings strengthen our study by providing rich insights into the participants' experiences and specifically in relation to SDT variables.

We do acknowledge that the majority of participants were female university students living with a physical disability or chronic condition. The LTPA outcomes may have been different for male students, university staff members, or adults living with other disabilities/impairments. Also, this study reports on two iterations of FAM, one iteration where the program was transitioned from in-person to online delivery due to the COVID-19 pandemic, and another iteration that was delivered online. The participants' experiences of the FAM program and LTPA may have been affected by the different program deliveries. Further research is warranted to explore how a need-supportive, adaptive LTPA environment may support LTPA participation and maintenance among a broader range of adults living with disabilities. Altogether, our study filled an important research gap regarding the experiences of individuals participating in an adapted physical activity program, and our findings can act as a stepping stone for future research in this area.

## 6. Conclusions

This study captured insights about the experiences and engagement in LTPA among university students and staff living with a physical disability or chronic condition upon completion of the adaptive physical activity program, FAM. This study found that if physical activity programs provide a need-supportive environment, university students and staff living with physical disabilities or chronic condition can experience positive support towards their psychological needs and increased levels of motivation and LTPA participation. Further research can build on this study to develop a more robust understanding as to how need-supportive, adapted LTPA programs could be refined and implemented within community settings.

**Supplementary Materials:** The following supporting information can be downloaded at: https://www.mdpi.com/article/10.3390/disabilities4020024/s1. Table S1. Sample Fitness Test/Assessment. Figure S1. Sample Exercise Prescription.

**Author Contributions:** T.M.L. led the qualitative methodology selection and design, qualitative data analysis, initial writing of the manuscript, revisions to the manuscript, and served as corresponding author, assuming responsibilities for submitting the manuscript and communicating revisions from the journal to co-authors. O.L.P. led the quantitative methodology selection and design, quantitative data analysis, initial writing of the manuscript, and revisions to the manuscript. G.D.B. provided support in the initial writing and revisions to the manuscript. C.C. provided support in the conceptualization of the study and participant recruitment. L.F. provided support in the conceptualization of the study and participant recruitment. R.D. provided support in the conceptualization of the study and participant recruitment. S.N.S. provided supervision, obtained and provided funding for the study, assisted with the conceptualization of the study, supported the qualitative and qualitative data analysis and contributed to revision of the manuscript. All authors have read and agreed to the published version of the manuscript.

**Funding:** This research was supported by McGill Social Sciences and Humanities Development grant.

**Institutional Review Board Statement:** The study was approved by McGill University Research Ethics Board (approval #111-0818, September, 2020).

**Informed Consent Statement:** Written consent was obtained by all participants involved in the study.

**Data Availability Statement:** The data are available upon request.

**Conflicts of Interest:** The authors declare no conflicts of interest.

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
