# Peer review of "Evaluating an Adapted Physical Activity Program for University Students and Staff Living with a Physical Disability and/or Chronic Condition through a Self-Determination Theory Lens"

_disabilities, doi:10.3390/disabilities4020024_

Round 1
Reviewer 1 Report
Comments and Suggestions for Authors
Dear authors
Thank you for submitting the manuscript entitled “Evaluating an Adapted Physical Activity Program for University Students and Staff Living with a Physical Disability and/or Chronic Condition Through a Self-Determination Theory Lens” for consideration for publication in Disabilities MDPI.
The paper is well-writing and provides evidence about the effect of a physical activity program on university personnel with physical disabilities or chronic diseases. The paper is well-written and uses a mixed design that combines quantitative and qualitative methodologies.
A few comments are explained below before making a final decision on its acceptance for publication in this journal:
ABSTRACT:
- Line 16. The introduction of the “FAM” abbreviation in the abstracts is confusing because it is not clear what the initials of the abbreviation mean.
- Line 17. I think the connector here is wrong: “…theory. Among university…”
- Line 32. I suggest removing “University” and “student” from the Keywords because they appear in the title of the manuscript. I also suggest using “physical exercise” instead of just “exercise”.
INTRODUCTION
- Line 39. What about “sensory” or “cognitive” impairments in “disability” definition”. Talking about “physical or psychological impairments” does not include the whole spectrum of disability.
- Line 56. Remove “and” between “2018” and “Williams et al., 2014”, and connect them with “;”.
- Line 60. Reference to the “WHO physical activity guidelines” should be included.
MATERIALS AND METHODS
- About your sample, did you conduct any sample size power analysis for your sample of 19 participants?
- Line 115. Remove the blank space between “FAM” and “program”.
- Line 116. Did you use a quantitative measurement after the three months of the end of the program for re-tests assessment?
- Line 125. Remove the blank space before “Upon”.
- Line 128. Appendixes A and B are not available for review.
- Line 151. The fact that you have had two groups because COVID-19 pandemic would be a positive contribution to the work, but this is not discussed properly in the discussion section. Specific conclusions would be obtained for both subgroups, on the contrary, this should be included as a limitation of the study.
- Lines 167-168. I suggest reporting the mean and SD of the interview time.
- Line 209. Did you conduct any normal distribution analyses (i.e., Kolmogorov-Smirnov or Levene tests) before your quantitative analyses?
- Line 213. Insert a blank space before “79-81”.
- Lines 236-238. How can you ensure that the “critical friend” acts without bias? Also, Is the “SS” abbreviation necessary? This does not appear again.
- Footnote of page 5: Please, report the percentage of those that have both a physical disability and a chronic disease.
RESULTS
- Table 1: report age as “M ± SD”, you are reporting them as “mean (SD)”
- Line 252. This line is confusing. Is this a subheading?
- Table 2. Please, homogenize all the decimals of the t and p values to the same number (i.e., 2 or 3).
- Line 261. I think the order of the date for the mils and moderate intensities is the contrary: i.e., 0.27 and 0.06, respectively.
- Lines 285-286: I suggest writing as follows: “For example, Jessie, a student in FAMP, explained that:”
- Lines 287-290. You are reporting the verbatim here in italics, but not in previous paragraphs. I suggest harmonizing all these transcriptions into the same format.
- Line 351. Remove the blank space after “community”.
- Line 356. I miss here a reference to “social” well-being because you discuss it later.
DISCUSSION, LIMITATIONS AND CONCLUSIONS
- Line 464: I suggest writing as “…disability and/or chronic…”.
- Line 478-479. Some basic information about the design of the program should be included in the manuscript. Without watching the supplementary material, the reader cannot understand that only ambulant persons can participate in the program. Did you consider wheelchair users or other support devices (e.g. crutches or aid walkers) as exclusion criteria.
- Line 610. Change “&” by “and” in the subheading.
- Line 650. Remove blank space before “The study…”
REFERENCES
- Line 667. Remove “:” after the title of the reference. Also, review the format of the pages of the references (one-page number is missing)
- Line 670. Does it is necessary to include a second year of the reference [i.e. “(2023)”]
- Line 680. Does it is necessary to include a second year of the reference [i.e. “(2021)”]
- Line 685. The title of the reference should be in italics.
- Line 687. Remove the blank space before the DOI.
- Line 699. DOI of the reference is incomplete.
- Line 702. “A” in upper case after “:”.
- Line 730. “A” in upper case after “:”.
- Line 736. Does it is necessary to include a second year of the reference [i.e. “(2002)”]. In this case, I think the year is wrong. Also. Insert a blank space before the number and issue.
- Line 749. The title of the reference is in Title Format. Review it to harmonize it with the rest of the references.
- Line 758. The city of the publication is not included.
- Line 761. Review the underlining of the DOI.
- Line 784. “A” in upper case after “:”.
- Lines 793-794. Review reference 53.
- Line 796. Remove the blank space between issue/numbers and pages of the reference.
Reviewer 2 Report
Comments and Suggestions for Authors
A mixed methods design was used to evaluate a campus-based adapted physical activity programs for university students and staff with physical disabilities and/or chronic conditions from a self-determination perspective. Nineteen participants received a 10-week program either on campus or online (during COVID-19). The program is clearly described. I have identified three major concerns that limit interpretation of the results.
My first concern is the reader has no context for understanding the results for the Godin Leisure Time Exercise Questionnaire (not cited in the references), Treatment Self-Regulation Questionnaire for Exercise, and the Psychological Need Satisfaction in Exercise Scale. Information is not provided on how scores are calculated and interpretation of scores for each measure. Table 2 lists nine variables but does not indicate what measure each variable is from. For example, the first variable is Total LTPA. What is the measure? What does a score of 184 represent? How is this score interpreted?
A second major concern is the interpretation of effect size. The authors’ state: “According to Cohen (1988, pp.79–81), the following guidelines were used to assess effect sizes and strength of the relationships: drm = 0.10 to 0.29 (small); drm = 0.30 to 0.49 (medium); drm = 0.50 to 1.0 (large).” Cohen proposed a d-index of .20 is a small effect, a d-index of .50 is a medium effect, and a d-index of ≥ .80 is a large effect. Consequently, a d-index of .00-.19 is insignificant, a d-index of .20-.49 is a small effect, and a d-index of .50-.79 is a medium effect (Portney & Watkins, 2009, p 649; Dunst & Hamby, 2012). Given the small sample and point estimate (confidence intervals for effect size estimates are not presented), reporting the percentage of non-overlap between pre-test and post-test scores associated with an effect size is recommended to assist readers in interpretation of the effect size statistic (i.e., What is a medium effect?).
A third concern is the qualitative phase. The interview questions are not provided. The authors’ state: “An interview guide was created for the interview, asking participants about their LTPA, autonomous and controlled motivation, and the basic psychological needs of SDT in the context of FAM, as well as their experiences engaging in the program.” The wording does not provide the reader with sufficient context for understanding the analysis and results.
The analysis and results of the interviews are not clear. On page 5 statements are made that “A directed content analysis was used to analyze the data, which uses prior theory to extend knowledge about a phenomenon (Hsieh & Shannon, 2005). Thus, co-authors TL and OP deductively coded the data based on SDT and the study variables. After familiarizing themselves with the data, TL and OP separately highlighted and coded the data using predetermined SDT categories where possible (Hsieh & Shannon, 2005).” Rather than identifying themes from analysis of interviews it appears that their responses were fit into pre-determined categories. Please elaborate on the rationale for this approach.
The themes identified from analysis of the interviews are not described. Rather selective quotes are presented following presentation of the effect size for each measure. On page 5 the authors’ state: “Next, TL and OP examined the categories to determine whether sub-themes were needed. After, coded data was organized into the predetermined SDT categories and coded text within established sub-themes were organized into the higher SDT categories. Text given another label were separately coded and organized into separate themes.” The process needs to be more clearly described. How were the predetermined categories identified? What are the higher SDY categories? What are the separate themes? A Table or examples are often useful in explaining the analysis process.
Round 2
Reviewer 1 Report
Comments and Suggestions for Authors
Thank for attending all the queries done in the first review.
Author Response
All revisions have been addressed in the first review.
Reviewer 2 Report
Comments and Suggestions for Authors
Thank you for responding to my comments. Issues remain for some of the concerns I identified in my original review. I have provided recommendations to address these issues.
Page 4 lines 194-197 The last two sentences of the paragraph require references.
The additional information provided for the three measures is useful; however, when reading Table 2 it remains unclear how scores for Competence, Relatedness, Autonomous Motivation, and Controlled Motivation were calculated. When describing each measure, indicate the variables measured in your study (Table 2) and how scores were calculated.
My previous concern that Table 2 lists nine variables but does not indicate what measure each variable is from was not addressed. A suggestion is to add a row to the Table with the name of each measure. For example, row one could state Godin Leisure Time Exercise Questionnaire informing the reader that the four variables for LTPA are from the Godin Leisure Time Exercise Questionnaire. This is especially important for the next five variables. Indicate the variables measured by the Treatment Self-Regulation Questionnaire for Exercise and the variables measured by the Psychological Need Satisfaction in Exercise Scale.
Thank you for the additional information on how the effect size was calculated. The information addresses my previous comment.
My previous comment that the interview questions are not provided has not been addressed. My point is that the wording is very technical (I would not expect participants to be familiar with terms such as autonomous and controlled motivation and self-determination theory). Please indicate the exact wording of the interview questions. A Table could be added that lists the interview questions.
The authors’ response addressed my comment on the rationale for directed content analysis. My recommendation is elaborate on the description on line 239 by incorporating the following from your response: “using a direct content analysis, which applies a deductive analysis, or pre-determine categories to organize and analyze qualitative data.”
The information that was added to Qualitative Analysis is useful. In their response the authors’ state: “The pre-determined and additional identified themes are the headers outlined in the results, section 2.2.1 – section 2.3.3.” Additionally, the authors state: “Given the mixed-methods approach of this study, both quantitative data (effect size) and qualitative data (interview quotes) were provided to demonstrate a measured effect and participants experiences associated with the predetermined and additional identified themes.” Please add these statements to the text. A suggestion is to place them at the beginning of the Results as follows: “Given the mixed-methods approach of this study, quantitative and qualitative results were integrated to demonstrate a measured effect and participants experiences associated with the predetermined and additional identified themes. The pre-determined and additional identified themes are the headers outlined in the results, section 2.2.1 – section 2.3.3.”
